# A Review of Partial Information Decomposition in Algorithmic Fairness and Explainability

**DOI:** 10.3390/e25050795

**Published:** 2023-05-13

**Authors:** Sanghamitra Dutta, Faisal Hamman

**Affiliations:** Department of Electrical and Computer Engineering, University of Maryland, College Park, MD 20742, USA; fhamman@umd.edu

**Keywords:** fairness, explainability, causality, information theory, partial information decomposition, unique information

## Abstract

Partial Information Decomposition (PID) is a body of work within information theory that allows one to quantify the information that several random variables provide about another random variable, either individually (unique information), redundantly (shared information), or only jointly (synergistic information). This review article aims to provide a survey of some recent and emerging applications of partial information decomposition in algorithmic fairness and explainability, which are of immense importance given the growing use of machine learning in high-stakes applications. For instance, PID, in conjunction with causality, has enabled the disentanglement of the non-exempt disparity which is the part of the overall disparity that is not due to critical job necessities. Similarly, in federated learning, PID has enabled the quantification of tradeoffs between local and global disparities. We introduce a taxonomy that highlights the role of PID in algorithmic fairness and explainability in three main avenues: (i) Quantifying the legally non-exempt disparity for auditing or training; (ii) Explaining contributions of various features or data points; and (iii) Formalizing tradeoffs among different disparities in federated learning. Lastly, we also review techniques for the estimation of PID measures, as well as discuss some challenges and future directions.

## 1. Introduction

Machine learning is being used in several high-stakes applications, such as hiring, education, finance, healthcare, etc., that directly impact and influence people’s lives. While these models are becoming very good at learning all patterns present in the data, blindly learning all patterns can sometimes have unintended consequences, such as propagating biases and stereotypes with respect to sensitive attributes, such as gender, race, age, nationality, etc., or taking decisions that we do not quite understand. Towards addressing these concerns, the fields of algorithmic fairness [1,2,3,4,5,6,7,8,9,10,11,12,13] and explainability [5,11,14,15,16,17,18,19] have received significant interest in recent times. In this review article, we explore fairness and explainability through the lens of an emerging body of work in information theory called Partial Information Decomposition (PID) [20,21,22,23,24].

Classical information-theoretic measures, such as mutual information, are widely used [9,25,26,27,28] in fairness to quantify the disparity (dependence) with respect to the sensitive attribute (*Z*) in the output of a model $\left(\hat{Y}\right)$. In several situations, however, only identifying disparity in the final output of a model (e.g., quantifying $I(Z;\hat{Y})$) is not enough. It becomes important to delve deeper and identify the information content that several random variables share about the sensitive attribute *Z*. Classical information-theoretic measures such as mutual information ($I(Z;\hat{Y})$) capture the entire dependency between *Z* and $\hat{Y}$ but fail to capture how this dependency is distributed among the input features, i.e., it says nothing about the structure of multivariate information [22]. Input features may contribute to ($I(Z;\hat{Y})$) in different ways. For instance, a feature (e.g., zip code) might provide unique information about *Z* (e.g., race) not arising from any other feature. Multiple features might also be providing the same (redundant) information about *Z* (e.g., zip code and county). Yet another interesting scenario arises if multiple features jointly provide information about *Z* not present in any feature individually (e.g., each individual digit of the zip code). To disentangle such joint information content between *Z* and $\hat{Y}$ among the contributing input features, we resort to a body of work in information theory called Partial Information Decomposition (PID). We discuss three motivational scenarios here:

### 1.1. Scenario 1: Quantifying Non-Exempt Disparity [8,29]

When it comes to legal disputes or designing policies for hiring, it becomes important to delve deeper and identify the sources of disparity, e.g., which attributes are responsible for the disparity and if those attributes are critical for that job. For instance, suppose one wants to hire a software engineer for a safety-critical application. An attribute such as coding test might be deemed very critical for the job whereas other attributes such as aptitude test might not be as critical. In fact, several existing discrimination laws (e.g. Title VII of the US Civil Rights Act [30]) allow exemptions if the disparity can be justified by an occupational necessity, i.e., “necessary to the normal operation of that particular business”, while prohibiting the remaining disparity. For example, a standardized coding-test score may be a critical feature in hiring software engineers for a safety-critical application. Similarly, weightlifting ability might be a critical feature in hiring firefighters so that they are able to carry victims out of a burning building. In these scenarios, it is important to quantify the legally *non-exempt disparity* which is the part of the disparity that is not due to the critical necessities.

### 1.2. Scenario 2: Explaining Contributions [31]

In many applications, e.g., college admissions, the decision-making mechanism can also be a complex combination of algorithms and human-in-the-loop. Thus, only identifying disparity in the final decision may not be enough to audit and, subsequently, mitigate them. E.g., there is an ongoing debate in the US on whether GRE/TOEFL scores should be used for college admissions because they may propagate disparity in the decisions with respect to sensitive attributes [32,33]. It is important to disentangle how the disparity in the decisions arose, e.g., which features could be potentially responsible for the disparity, and then evaluate how critical those features are for the specific application.

### 1.3. Scenario 3: Formalizing Tradeoffs in Distributed Environments [34]

Similarly, in distributed machine learning and federated learning, if the global population demographics are quite different from the local demographics where the model is being deployed, it has been observed that the “local” and “global” notions of disparity (dependence) can be quite different, leading to widespread concerns. In this scenario, it is critical to formalize the tradeoffs between “local” and “global” disparity, by examining the joint information content about the sensitive attribute in the model output and the local demographics.

In all of these scenarios, Partial Information Decomposition (PID) provides a framework [20,21,22,23,24] for characterizing the joint information content that several random variables contain about one random variable (often referred to as the message), either individually (unique information), redundantly (shared information), or only jointly (synergistic information). In Scenario 1, PID has been found to be particularly useful in quantifying the legally non-exempt disparity which is the part of the disparity that is not due to the critical necessities. In Scenario 2, PID has enabled the disentanglement of the contribution of different features or different data points to the overall disparity. In Scenario 3, PID has enabled a formal quantification of the interplay between “local” and “global” disparities.

Partial Information Decomposition (PID) bridges the fields of fairness, explainability, and information theory. In this review article, we aim to provide a unified survey of some recent and emerging applications of PID in this area. We introduce a taxonomy that highlights the role of Partial Information Decomposition (PID) in the three main avenues: (i) Quantifying the legally non-exempt disparity for auditing or training; (ii) Explaining contributions of various features or data points; and (iii) Formalizing tradeoffs among different disparities in federated learning. Lastly, we also review several techniques for the estimation of PID measures, as well as discussing some challenges and future directions.

This review article is organized as follows: In Section 2, we provide a brief background on Partial Information Decomposition. In Section 3, we first introduce the problem setup for quantifying non-exempt disparity as discussed in [8,29], and present several canonical examples and candidate measures, understanding their pros and cons, until we arrive at the proposed measure in [8] that satisfies the desirable properties. In Section 4, we review how PID can help in assessing the contributions of either features or data points with applications in feature selection (as discussed in [31]). Related works include [35,36,37,38]. In Section 5, we discuss another avenue where PID plays an important role: quantifying tradeoffs between different measures, as we illustrate through the example of local and global fairness in federated learning (as discussed in [34]). We conclude with a discussion on estimation techniques for the PID measures and future directions in Section 6.

## 2. Background on Partial Information Decomposition

The PID framework [20,22,39] decomposes the mutual information I(Z;(A,B)) about a random variable *Z* contained in the tuple (A,B) into four *non-negative* terms as follows (also see Figure 1):(1)$$\begin{array}{cc} I(Z;(A,B)) & =\mathrm{Uni}(Z:A|B)+\mathrm{Uni}(Z:B|A)+\mathrm{Red}(Z:(A,B))+\mathrm{Syn}(Z:(A,B)). \end{array}$$

Here, Uni(Z:A|B) denotes the unique information about *Z* that is present only in *A* and not in *B*. Likewise, Uni(Z:B|A) is the unique information about *Z* that is present only in *B* and not in *A*. The term Red(Z:(A,B)) denotes the redundant information about *Z* that is present in both *A* and *B*, and Syn(Z:(A,B)) denotes the synergistic information not present in either of *A* or *B* individually, but present jointly in (A,B). Before defining these PID terms formally, let us understand them through an intuitive scenario.

**Example** **1**(Understanding Partial Information Decomposition)**.**
*Let $Z=(Z_{1},Z_{2},Z_{3})$ with $Z_{1},Z_{2},Z_{3}∼$ i.i.d. Bern(½). Let $A=(Z_{1},Z_{2},Z_{3}⊕N)$, $B=(Z_{2},N)$, N∼ Bern(½) is independent of Z. Here, I(Z;(A,B))=3 bits.*

The unique information about *Z* that is contained only in *A* and not in *B* is effectively contained in the variable $Z_{1}$ and is given by $\mathrm{Uni}\left(Z:A|B\right)=I\left(Z;Z_{1}\right)=1$ bit. The redundant information about *Z* that is contained in both *A* and *B* is effectively contained in $Z_{2}$ and is given by $\mathrm{Red}\left(Z:\left(A,B\right)\right)=I\left(Z;Z_{2}\right)=1$ bit. Lastly, the synergistic information about *Z* that is not contained in either *A* or *B* alone, but is contained in both of them together is effectively contained in the tuple $\left(Z_{3}⊕N,N\right)$, and is given by $\mathrm{Syn}\left(Z:(A,B)\right)=I\left(Z;(Z_{3}⊕N,N)\right)=1$ bit. This accounts for the three bits in I(Z;(A,B)). Here, *B* does not have any unique information about *Z* that is not contained in *A*, i.e., Uni(Z:B|A)=0.

Irrespective of the formal definitions, the following identities also hold (see Figure 1): (2)$$\begin{array}{cc} \; & I(Z;A)=\mathrm{Uni}(Z:A|B)+\mathrm{Red}(Z:(A,B)). \end{array}$$(3)$$\begin{array}{cc} \; & I(Z;A|B)=\mathrm{Uni}(Z:A|B)+\mathrm{Syn}(Z:(A,B)). \end{array}$$

**Remark** **1**(Interpretation of PID as Information-Theoretic Sub-Volumes)**.**
*Uni(Z:A|B) can be viewed as the information-theoretic sub-volume of the intersection between I(Z;A) and I(Z;A∣B). Similarly, Red(Z:(A,B)) is the sub-volume between I(Z;A) and I(Z;B).*

Given three independent Equations (Equation 1)–(Equation 3) in four unknowns (the four PID terms), defining any one of the terms (e.g., Uni(Z:A|B)) is sufficient to obtain the other three. For completeness, we include the definition of unique information from [20] (that also allows for estimation via convex optimization [40]). To follow the paper, an intuitive understanding is sufficient.

**Definition** **1**(Unique Information [20])**.**
*Let* Δ *be the set of all joint distributions on (Z,A,B) and $\Delta_{p}$ be the set of joint distributions with the same marginals on (Z,A) and (Z,B) as their true distribution, i.e., $\Delta_{p}=\left\{Q\in \Delta :q\left(z,a\right)=\mathrm{Pr}\left(Z=z,A=a\right)\;\;\mathrm{and}\;\;q\left(z,b\right)=\mathrm{Pr}\left(Z=z,B=b\right)\right\}.$ Then, $\mathrm{Uni}\left(Z:A|B\right)=\min_{Q\in \Delta_{p}}I_{Q}\left(Z;A|B\right),$ where $I_{Q}\left(Z;A|B\right)$ is the conditional mutual information when (Z,A,B) have joint distribution Q.*

The key intuition behind this definition is that the unique information should only depend on the marginal distribution of the pairs (Z,A) and (Z,B). This is motivated from an operational perspective that, if *A* has unique information about *Z* (with respect to *B*), then there must be a situation where one can predict *Z* better using *A* than *B* (more details in ([20] Section 2)). Therefore, all the joint distributions in the set $\Delta_{p}$ with the same marginals essentially have the same unique information, and the distribution $Q^{*}$ that minimizes $I_{Q}\left(Z;A|B\right)$ is the joint distribution that has no synergistic information leading to $I_{Q^{*}}\left(Z;A|B\right)=\mathrm{Uni}\left(Z:A|B\right)$. Definition 1 also helps us define Red(Z:(A,B)) and Syn(Z:(A,B)) using (Equation 2) and (Equation 3). For discrete random variables, one can also refer to the package [41] to learn more about the different definitions of PID measures as well as different techniques of computing them.

With this brief background on PID, we now move onto discussing its role in quantifying non-exempt disparity.

## 3. Quantifying Non-Exempt Disparity

### 3.1. Preliminaries

A notable problem in algorithmic fairness (as discussed in [3,8,29,42,43,44,45,46,47]) is to check if the disparity in a model arose purely due to the critical features or not, e.g., coding skills for hiring a software engineer for a safety-critical application. Let *X* denote the input features which consist of the critical features $X_{c}$ and general features $X_{g}$. The model produces output $\hat{Y}$ which is a deterministic function of *X*. Consistent with several other works on fairness [45,48,49], these features are assumed to be generated from an underlying structural causal model (see Definition 2) where the latent variables *U* represent possibly unknown social factors. The observables *V* consist of the protected attributes *Z*, the features *X* and the output $\hat{Y}$. For simplicity, refs. [8,29] assumes ancestral closure of the protected attributes, i.e., the parents of any $V_{i}\in Z$ also lie in *Z*. For completeness, the definition of a structural causal model (SCM) is included here (also see Figure 2 for more details).

**Definition** **2**(Structural Causal Model: SCM(U,V,F) [50])**.**
*A structural causal model (U,V,F) consists of a set of latent (unobserved) and mutually independent variables U which are not caused by any variable in the set of observable variables V, and a collection of deterministic functions (structural assignments) $F=(F_{1},F_{2},\ldots )$, one for each $V_{i}\in V$, such that: $V_{i}=F_{i}\left(V_{pa_{i}},U_{i}\right).$ Here $V_{pa_{i}}\subseteq V\backslash V_{i}$ are the parents of $V_{i}$, and $U_{i}\subseteq U$. The structural assignment graph of SCM(U,V,F) has one vertex for each $V_{i}$, and directed edges to $V_{i}$ from each parent in $V_{pa_{i}}$, and is always a directed acyclic graph.*

### 3.2. Quantifying Non-Exempt Disparity

Towards quantifying non-exempt disparity, we discuss several canonical examples and desirable properties that help us arrive at a measure of non-exempt disparity. We start with a brief discussion on two popular metrics of fairness (see some popular metrics and their implementations in [51]), namely, statistical parity and equalized odds, which have no provision for selective quantification of disparity due to specific features.

Statistical parity suggests that a model is fair if the output $\hat{Y}$ is entirely independent of *Z*. There are several ways to incorporate statistical parity [51], such as minimizing the absolute gap $|\mathrm{Pr}\left(\hat{Y}|Z=1\right)-\mathrm{Pr}(\hat{Y}|Z=0)|$ either during pre-processing, training, or post-processing. An information-theoretic measure of statistical parity is mutual information $I(Z;\hat{Y})$ which goes to zero if and only if $\hat{Y}$ is entirely independent of *Z*. However, statistical parity has no provision for selectively quantifying the part of the disparity which is due to the critical features $X_{c}$.

On the other hand, equalized odds suggest that a model is fair if the output $\hat{Y}$ is independent of *Z* conditioned on the true label *Y*. There are several ways to incorporate equalized odds [51] as well, such as minimizing the absolute gap $|\mathrm{Pr}\left(\hat{Y}|Z=1,Y=1\right)-\mathrm{Pr}(\hat{Y}|Z=0,Y=1)|$ either during pre-processing, training, or post-processing. The equalized odds condition becomes information-theoretically equivalent [28] to setting the conditional mutual information zero, i.e., $I(Z;\hat{Y}|X_{c})=0$. The rationale is that conditioning on the true labels might help in exempting the correlation with *Z* that is already present in the true label *Y*. However, equalized odds also have some limitations, particularly when there is historic bias in the true labels themselves (further discussed in [8]). The problem of quantifying non-exempt disparity adopts a middle ground between statistical parity (no exemption at all due to critical necessities) and equalized odds (exempts all disparities in past labels even if labels are biased): the goal is to selectively quantify the non-exempt disparity which is the part of the disparity that is not due to the critical features $X_{c}$.

To address the limitations of both statistical parity and equalized odds in appropriately capturing non-exempt disparity, another candidate measure that has been considered is conditional mutual information $I(Z;\hat{Y}|X_{c})$ (also referred to as conditional statistical parity [43]). The rationale is that conditioning on the critical feature $X_{c}$ might help in exempting the correlation with *Z* already present in $X_{c}$. For instance, if $Z-X_{c}-\hat{Y}$ forms a Markov chain, then the conditional mutual information $I(Z;\hat{Y}|X_{c})$ would go to zero.

However, refs. [8,29] make a critical observation: conditioning on $X_{c}$ naively can sometimes also capture misleading dependencies (or correlations) even if the model output happens to be causally fair (and independent of *Z*). We illustrate this issue with the following counterexample (also see Figure 3 (Left)).

**Counterexample** **1**(Counterfactually Fair Hiring). *Let Z∼ Bern(½) be the sensitive attribute, U∼ Bern(½) be the inner ability of a candidate, and $X_{c}=\left\{\begin{array}{cc} U, & Z=0\\ U+1, & Z=1 \end{array}\right.$ be the coding-test score (critical feature). This can be rewritten as $X_{c}=Z\left(U+1\right)+\left(1-Z\right)U=Z+U.$ However, instead of only using the biased test score, suppose the company chooses to conduct a thorough evaluation of their online code samples, leading to another score that distills out their inner ability, i.e., $X_{g}=U$. Suppose the model for hiring that maximizes accuracy turns out to be $\hat{Y}=X_{g}=U$.*

Notice that this model is deemed *fair* by causal definitions of fair (e.g., counterfactual fairness) because the output $\hat{Y}$ has no causal influence of *Z* (no causal path from *Z* to $\hat{Y}$). Even though the disparity from $X_{c}$ is legally exempt, the trained black-box model happens to base its decisions on another available non-critical/general feature that has no causal influence of *Z*. Thus, there is no disparity in the outcome $\hat{Y}$ (this is true even if the features in $X_{c}$ were not exempt). Therefore, it is desirable that the non-exempt disparity also be 0. However, the candidate measure $I\left(Z;\hat{Y}|X_{c}\right)=I\left(Z;U|Z+U\right)$ is non-zero here, leading to a false positive conclusion in detecting non-exempt disparity. Thus, it is desirable that *a measure of non-exempt disparity should go to zero whenever a model is causally fair.*

It is this limitation of conditional mutual information $I(Z;\hat{Y}|X_{c})$ that leads [8] to delve into Partial Information Decomposition which further decomposes $I(Z;\hat{Y}|X_{c})$ into two terms: Unique Information $\mathrm{Uni}(Z:\hat{Y}|X_{c})$ and Synergistic Information $\mathrm{Syn}(Z:\hat{Y},X_{c})$. It has been demonstrated that the Unique Information $\mathrm{Uni}(Z:\hat{Y}|X_{c})$ satisfies the desirable property stated above and also resolves Counterexample 1. We also include a comparison of these measures in Table 1.

### 3.3. Demystifying Unique Information as a Measure of Non-Exempt Disparity

We note that mutual information $I(Z;\hat{Y})$ captures the entire statistical disparity (dependence) between the protected attribute *Z* and the model output $\hat{Y}$, irrespective of which feature it is arising from (see Figure 3 (Right)). So, essentially $I(Z;\hat{Y})$ does not allow for any exemptions due to critical necessities. On the other hand, $I(Z;\hat{Y}|X_{c})$ attempts to exempt some of the disparity that is only due to the critical necessities, but it also ends up capturing additional dependencies even when $I(Z;\hat{Y})=0$ (refer to the Venn diagram representation in Figure 3; such a scenario was captured in Counterexample 1). Thus, Unique Information $\mathrm{Uni}(Z:\hat{Y}|X_{c})$ is the proposed measure of non-exempt disparity because it captures the intersection between mutual information $I(Z;\hat{Y})$ and conditional mutual information $\mathrm{Uni}(Z:\hat{Y}|X_{c})$. The unique information $\mathrm{Uni}(Z:\hat{Y}|X_{c})$ satisfies several desirable properties (including several monotonicity properties):

**Theorem** **1**(Properties of Unique Information)**.**
*Unique information $\mathrm{Uni}(Z:\hat{Y}|X_{c})$ satisfies several desirable properties of a measure of non-exempt disparity as follows:*
*$\mathrm{Uni}(Z:\hat{Y}|X_{c})=0$ if the model is causally fair.**$\mathrm{Uni}\left(Z:\hat{Y}|X_{c}\right)=I\left(Z;\hat{Y}\right)$ if all features are non-critical, i.e., $X_{c}=\phi$ and $X_{g}=X$.**For a fixed set of features X and a fixed model $\hat{Y}=h\left(X\right)$, a $\mathrm{Uni}(Z:\hat{Y}|X_{c})$ should be non-increasing if a feature is removed from $X_{g}$ and added to $X_{c}$.**$\mathrm{Uni}(Z:\hat{Y}|X_{c})=0$ if all features are critical, i.e., $X_{c}=X$ and $X_{g}=\phi$.*

This result is a simplified adaptation from [8] which also contains the proof.

Not only does unique information $\mathrm{Uni}(Z:\hat{Y}|X_{c})$ allow for auditing models for non-exempt disparity, but it can also be used to selectively minimize non-exempt disparity if desired. In [8], different information-theoretic measures are incorporated as regularizers with the loss function to selectively reduce non-exempt disparity if desired.

While the unique information $\mathrm{Uni}(Z:\hat{Y}|X_{c})$ satisfies several desirable properties as a measure of non-exempt disparity, ref. [8] goes on to explore more nuanced examples involving non-faithful structural causal models where no purely observational measure would successfully capture all the desirable properties, leading to novel measures that bridge causality and partial information decomposition. In particular, the proposed measure is given by: $$M_{NE}^{*}=\min_{U_{a},U_{b}}\mathrm{Uni}\left(\left(Z,U_{a}\right):\left(\hat{Y},U_{b}\right)|X_{c}\right)\mathrm{such}\;\mathrm{that}\;U_{a}=U_{X}\backslash U_{b},$$
where the minimization is over all possible partitioning of the set of latent random variables *U*. We refer interested readers to [8] for more details, as well as more counterexamples that contrast the proposed measures from other purely causal path-based approaches.

## 4. Explaining Contributions

### 4.1. Preliminaries

In many applications, e.g., college admissions, the decision-making mechanism can also be a complex combination of algorithms and human-in-the-loop. The final decision is denoted by $\hat{Y}$ which is a complex combination of the deterministic model output h(X) and subjective evaluation by human-in-the-loop who may take additional factors (non-quantified aspects) into consideration. These additional factors almost always include the protected attribute *Z*, e.g., gender, race, age, etc. Therefore, the final decision $\hat{Y}$ may not be a deterministic function of the model inputs *X*, and could also depend on *Z*. In this scenario, a notable problem of interest is: *how to quantify the contribution of each individual feature to the overall observed disparity $I(Z;\hat{Y})$?*

### 4.2. Information-Theoretic Measures

Towards answering this question, ref. [31] proposes two measures for quantifying the contribution of each feature to the overall disparity. The first measure, which is referred to as *interventional contribution*, is defined as follows:$$\mathrm{Contri}\left(X_{i}\right)=\sum_{X_{S}\subseteq X\backslash X_{i}}\frac{|X_{S}\left|!\;(n-\right|X_{S}\left|-1\right)!}{n!}\left(I\left(Z;\hat{Y}\left(X_{S}\cup X_{i}\right)\right)-I\left(Z;\hat{Y}\left(X_{S}\right)\right)\right).$$
Here, $\hat{Y}\left(X_{S}\right)$ denotes the output of the model/decision-making system using the same model but with only the features in the set $X_{S}\subseteq X$ (often, the other inputs are set as constants). This measure quantifies the contribution of each individual feature to the overall disparity in a Shapley-value-inspired manner. While this measure satisfies several desirable properties, such as the contributions being non-negative and adding up to $I(Z;\hat{Y})$ (just like Shapley values), there are certain scenarios, e.g., feature selection where one might be more interested in quantifying “potential” rather than “interventional” contribution. This is because there may be two features that are quite strongly correlated, and yet only one of them might actually be used by the model (e.g., $\hat{Y}=X_{1}$ and $X_{1}=X_{2}$). So, $\mathrm{Contri}(X_{1})$ would be high while $\mathrm{Contri}(X_{2})$ may be 0. However, if one were to drop $X_{1}$ and retrain a new model using only $X_{2}$, the disparity would not be removed since $X_{2}$ essentially encodes the same biases and stereotypes as $X_{1}$. Thus, an alternate measure of quantifying contribution, which is referred to as *potential contribution*, is defined as follows:$$\mathrm{PotentContri}\left(X_{i}\right)=\;\;\;\;\sum_{X_{S}\subseteq X\backslash X_{i}}\;\;\;\;\;\frac{|X_{S}\left|!\;(n-\right|X_{S}\left|-1\right)!}{n!}\left(\mathrm{Red}\left(Z:(\hat{Y},X_{S}\cup X_{i})\right)-\mathrm{Red}\left(Z:(\hat{Y},X_{S})\right)\right).$$

Closely connected is the literature on explainability [15,52] (also see [18] for a survey). Broadly speaking, the goal of explainability techniques, such as SHAP [15] is to quantify the contribution of each individual feature to the decision $\hat{Y}$ *locally around a specific point*. There have been extensions of SHAP to quantify feature contributions to statistical parity, by adding the feature contributions separately for data points corresponding to different protected groups. Instead, ref. [31] focuses on introducing information-theoretic measures to examine the problem from a distributional lens, and contrasting *contribution* and *potential contribution*, also touching upon the issue of substitute features. We include a summary in Table 2. We also refer to [31] for a more detailed discussion.

### 4.3. Notable Related Works Bridging Fairness, Explainability, and Information Theory

A closely related direction of research that bridges fairness and explainability is the problem of feature selection for algorithmic fairness [35,36,37,53]. In [35,37], the authors propose novel information-theoretic techniques that leverage conditional mutual information with the goal of selecting a subset of features that would achieve fairness, in particular, *justifiable fairness* [47]. In [36], the authors explore the problem of feature selection for algorithmic fairness and propose techniques that leverage partial information decomposition to achieve an improved tradeoff between fairness and accuracy.

In [38], authors introduce the notion of “unique sample information”, which captures the contribution that a particular sample provides to the training of a neural network. In other words, it quantifies the information that a sample provides to the weights. They define this measure as the KL divergence between the distribution of the weights of the network trained with and without the sample. The paper provides efficient methods to compute unique sample information and demonstrates its applications in various problems, such as analyzing the informativeness of different data sources and detecting adversarial and corrupted samples.

## 5. Formalizing Tradeoffs in Distributed Environments

Next, we discuss yet another important application of PID, i.e., formalizing fairness tradeoffs in distributed environments, e.g., federated learning. Federated learning (FL) is a framework that allows multiple parties, commonly referred to as clients, to collectively train machine learning models while preserving the privacy of their local data [54]. However, due to the decentralized nature of data in the FL setting, group fairness analysis becomes a significant challenge.

Existing literature on group fairness [55,56,57,58] in FL has highlighted two main forms of fairness: global and local fairness. Global fairness pertains to the disparity of the developed model when evaluated on the entire dataset across all clients. For instance, in a scenario where several banks engage in FL to train a model for determining loan qualifications, a globally fair model is one that does not discriminate against any protected group when evaluated on the complete dataset across all the banks. However, achieving global fairness is non-trivial since each client only has access to their own dataset. On the other hand, local fairness pertains to the disparity of the model at each individual client, i.e., when evaluated on a client’s local dataset.

One might notice that global [55,56,57] and local fairness evaluation can differ from each other when the local demographics at a client differ from the global demographics across the entire dataset (data heterogeneity, e.g., a bank with customers predominantly of a particular race). Previous research has mostly focused on trying to achieve global fairness [55,56,57] without specifically considering its interplay with local fairness [58]. There is a lack of understanding of the relationship between these two concepts, and if and when, one implies the other. When the data are i.i.d. across clients, it is generally understood that global and local fairness would be the same, but their interplay in other situations is not well understood. In this context, PID provides a tool for breaking down global and local disparities into various components, which in turn reveals the fundamental information-theoretic limits and trade-offs that exist between these disparities [34].

### 5.1. Preliminaries

We let *S* denote the client, *X* the input features, *Z* the protected attribute, and *Y* the true label. The model *f* produces output $\hat{Y}$ which is a deterministic function of *X*. The global disparity of a model *f* with respect to *Z* can be measured as the mutual information between *Z* and $\hat{Y}$, denoted by $I(Z,\hat{Y})$. This notion aligns with the statistical parity definition of group fairness, which suggests that a model is fair if the predicted output $\hat{Y}$ is independent of *Z*. On the other hand, local fairness is essentially the statistical parity evaluated at each local client, i.e., $Z⫫\hat{Y}$ given each S=s. Thus, ref. [34] defines the local disparity as the conditional mutual information between *Z* and $\hat{Y}$ conditioned on *S*, i.e., $I(Z;\hat{Y}|S)$.

### 5.2. Partial Information Decomposition of Disparity in FL

Using PID, ref. [34] demonstrates fundamental limitations and tradeoffs between local and global disparity. The global and local disparity can be decomposed using PID as follows: (4)$$\begin{array}{cc} \; & I\left(Z;\hat{Y}\right)=\mathrm{Uni}\left(Z:\hat{Y}|S\right)+\mathrm{Red}\left(Z:\hat{Y},S\right). \end{array}$$(5)$$\begin{array}{cc} \; & I\left(Z;\hat{Y}|S\right)=\mathrm{Uni}\left(Z:\hat{Y}|S\right)+\mathrm{Syn}\left(Z:(\hat{Y},S)\right). \end{array}$$

The *Unique Disparity* $\mathrm{Uni}(Z:\hat{Y}|S)$ represents the information about *Z* that is exclusively present in the model prediction $\hat{Y}$ but not in the client *S*. The *Redundant Disparity* $\mathrm{Red}(Z:\hat{Y},S)$ denotes the overlapping information about *Z* that is present in both $\hat{Y}$ and *S*. Finally, the *Masked Disparity*
$\mathrm{Syn}(Z:(\hat{Y},S))$ reflects the synergistic information about *Z* that is only observed when $\hat{Y}$ and *S* are considered jointly and not present in either $\hat{Y}$ or *S* individually. Refer to Figure 4 for a graphical representation.

Consider a FL setting with two clients, where the protected attribute is binary (men and women), and the model predictions are also binary. We first examine three canonical examples of three corresponding types of disparities.

Pure Uniqueness: $\hat{Y}=Z$ and Z⫫S. The model assigns a positive prediction to men from each client dataset and makes its predictions based solely on the sensitive attribute. The unique disparity $\mathrm{Uni}(Z,\hat{Y}|S)=1$ with zero redundant and masked disparity since all the information about the protected attribute *Z* is encoded in the model predictions $\hat{Y}$, and none is present in client *S*. Such a model is both locally and globally unfair.

Pure Redundancy: $\hat{Y}=Z=S$. The protected attributes are skewed across clients, and the model makes its predictions based on both the protected attribute *Z* and the client *S*. The redundant disparity $\mathrm{Red}(Z;\hat{Y},S)=1$, with zero unique and mask disparity since all the information about the protected attribute *Z* is present in both the model predictions $\hat{Y}$ and the client *S*. This model achieves local fairness but is globally unfair. In general, pure redundant disparity is observed when $Z-S-\hat{Y}$ form a *Markov chain*, but *Z* and $\hat{Y}$ are correlated, i.e., $\hat{Y}=S$ and S=g(Z) for some function *g*.

Pure Synergy: $\hat{Y}=Z⊕S$ and Z⫫S. The model predictions are an XOR of the sensitive attribute and the client attribute, i.e., the model assigns positive prediction to men from client S=0 and women from client S=1, while all others are assigned a negative prediction. The model achieves global fairness by balancing the local unfairness at each client. There is zero unique and redundant disparity in this model because neither the model predictions $\hat{Y}$ nor the client *S* contain any information about the protected attribute. However, the masked disparity $\mathrm{Syn}(Z;\left(\hat{Y},S\right))=1$, since the information about the protected attribute *Z* is only present when both the model predictions $\hat{Y}$ and the client *S* are considered jointly.

### 5.3. Fundamental Limits and Tradeoffs between Local and Global Fairness

First, ref. [34] formally shows that, even if the local clients are able to reduce the local disparity to zero, the global disparity may still be non-zero. This is due to the redundant disparity and can be visualized using the Venn diagram in Figure 4. This has practical implications for deploying locally fair models, because even using optimal local mitigation and model aggregation techniques may not eliminate global disparity if the redundant disparity is present.

Similarly, ref. [34] shows that even if the global disparity is reduced to zero, the local disparity may still be non-zero due to the masked disparity. This can be seen pictorially in Figure 4, where it is evident that reducing the global disparity only decreases the redundant and unique disparities, but not the masked disparity. In other words, this means that although we may be able to train a model to achieve global fairness, it may not translate to fairness at the local client. For more details and experimental results, we refer to [34].

Thus, it is crucial to consider the presence of unique, redundant, and masked disparity when attempting to achieve global or local fairness. PID provides a framework for quantifying these different types of disparity, allowing for a more nuanced understanding of the tradeoffs and limitations involved in achieving fairness in FL. This understanding can inform the use of disparity mitigation techniques, their convergence, and the effectiveness of models when deployed in practice.

## 6. Discussion

Lastly, we conclude with a brief discussion on estimation techniques for Partial Information Decomposition (PID) and some future research directions.

### 6.1. Estimation of PID Measures

The field of Partial Information Decomposition (PID) has seen growing interest, with several PID measures being proposed, as well as several approaches to estimate them. Building on the original proposition in [22] which looks at the minimum specific mutual information, in [59] the focus is on defining an alternate measure of redundant information from the perspective of common information. In [20,39], the unique information component of the decomposition turns out to be the minimum value of the conditional mutual information over a constrained set of information channels. While this definition has an operational interpretation, it is only defined for the bivariate case, i.e., when the joint information about *Z* in two random variables (A,B) are of interest. Ref. [40] presents an efficient iterative divergence minimization algorithm to solve this optimization problem with convergence guarantees and evaluate its performance against other techniques. Ref. [60] proposes a general framework for constructing a multivariate PID, leveraging set theory, to define a PID in terms of the Blackwell order. Ref. [61] defines a measure of redundant information based on projections in the space of probability distributions. Ref. [62] presents a new measure of redundancy that quantifies the common change in surprise shared between variables at the local or point-wise level. Ref. [63] proposes a measure for unique information based on the dependency decomposition method that delineates how statistical dependencies influence the structure of a joint distribution. Ref. [64] uses an approach using specificity and ambiguity lattices. Ref. [24] proposes a novel quantification of PID using Markov relations.

There are estimation challenges for information-theoretic measures (see [65,66] and the references therein). Designing estimators building upon techniques proposed in [25,27,66,67] is an interesting direction of research. In [68], a method for estimating the unique information for continuous distributions is proposed. Their method solves the associated optimization problem over the space of distributions with fixed bivariate marginals by combining copula decompositions and techniques developed to optimize variational autoencoders. In [69], a method that enables the approximation of the redundant information that high-dimensional sources contain about a target variable.

### 6.2. Summary of Contributions

In essence, Partial Information Decomposition (PID) provides a valuable tool to understand and decompose the information content that several random variables contain about another random variable, either uniquely, redundantly, or synergistically. It plays an important role in trustworthy machine learning, particularly at the intersection of fairness and explainability, which are of immense importance given the growing use of machine learning in high-stakes applications. In this review paper, we focus on three scenarios where PID is indispensable: (i) Quantifying the legally non-exempt disparity for auditing or training; (ii) Explaining contributions of various features or data points; and (iii) Formalizing tradeoffs among different disparities in federated learning. PID holds the potential to provide a unified perspective on fairness and explainability, leading to several interesting future research problems, including understanding the patterns that a model learns more generally [68] and questioning when a model learns a misleading or spurious correlation. Closely related is its interplay with causal inference and representation learning [70] as well as its role in understanding fundamental information-theoretic tradeoffs [10,34] in trustworthy machine learning more broadly.

## Figures and Tables

**Figure 1 entropy-25-00795-f001:**
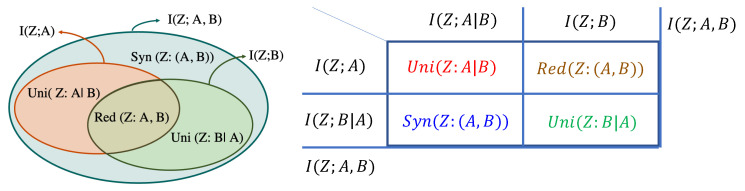
Illustration of PID: (**Left**) Venn diagram showing the Partial Information Decomposition of I(Z;(A,B)). (**Right**) Tabular representation of PID to help understand Equations (Equation 1)–(Equation 3).

**Figure 2 entropy-25-00795-f002:**
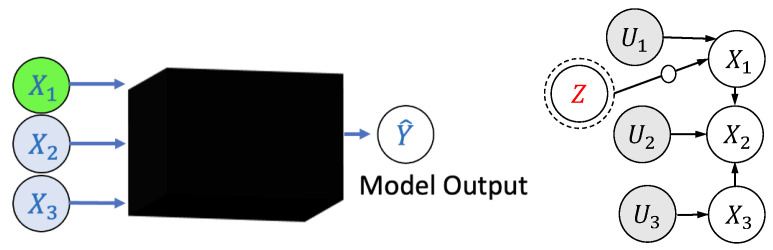
(**Left**) Machine learning model taking features $\left(X_{1},X_{2},X_{3}\right)$ as input and producing $\hat{Y}$ as output. (**Right**) The structural causal model denotes the underlying data generation process. Here, *U*s denote unobserved latent random variables that are independent, and *Z* is the sensitive attribute.

**Figure 3 entropy-25-00795-f003:**
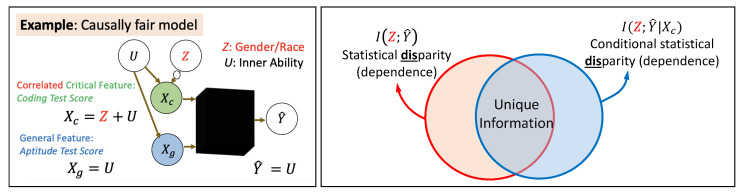
(**Left**) Counterexample 1: $I(Z;\hat{Y}|X_{c})>0$ even when model is causally fair. (**Right**) Demystifying unique information as a measure of non-exempt disparity.

**Figure 4 entropy-25-00795-f004:**
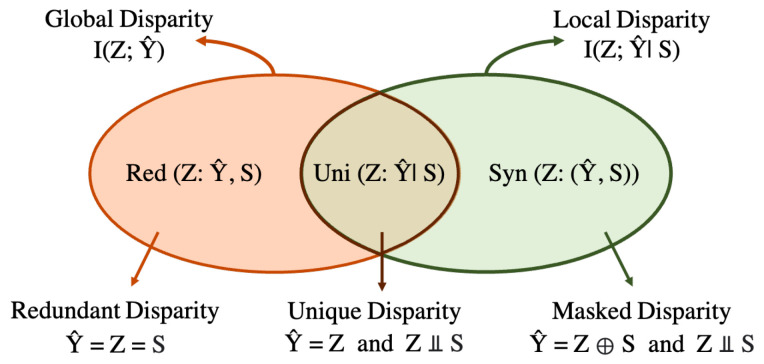
Venn diagram showing PID of Global and Local Disparity with canonical examples where each disparity is maximum [34].

**Table 1 entropy-25-00795-t001:** Observational fairness measures in the context of quantifying non-exempt disparity.

Measure	Discussion
Statistical Parity ($I(Z;\hat{Y})$)	Quantifies entire dependence between *Z* and $\hat{Y}$ with no exemptions. Does not quantify feature-specific contribution to disparity.
Equalized Odds ($I(Z;\hat{Y}|Y)$)	Quantifies dependence between *Z* and $\hat{Y}$ conditioned on the past labels *Y*. No feature-specific contribution to disparity; also may suffer from label bias.
Conditional Statistical Parity ($I(Z;\hat{Y}|X_{c})$)	Dependence between *Z* and $\hat{Y}$ conditioned on critical feature $X_{c}$ to allow exemptions. May sometimes be non-zero even when $\hat{Y}$ is independent of *Z* (or even when $\hat{Y}$ has no causal influence of *Z*).
Unique Information ($\mathrm{Uni}(Z:\hat{Y}|X_{c})$)	Unique dependence between *Z* and $\hat{Y}$ arising only due to $X_{c}$, satisfying desirable properties (Theorem 1). Misses masked disparities for which causality may be required (see [8] for a measure bridging PID and causality).

**Table 2 entropy-25-00795-t002:** Explainability measures in the context of quantifying contribution to disparity.

Measure	Discussion
SHAP [15] (Can be adapted for disparity)	Local explainability technique to obtain the contributions of features to the output of a given model around a point. Does not account for redundant features that can substitute an important feature when it is dropped.
Interventional Contribution to disparity	Global explainability technique to obtain the contributions of features to the output of a given model. Does not account for redundant features that can substitute an important feature when it is dropped.
Potential Contribution to disparity	Global explainability technique that is specifically tailored towards potential contribution towards disparity. Accounts for redundant features that can substitute an important feature when it is dropped.

## Data Availability

Not applicable.

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
