# Peer review of "A Review of Partial Information Decomposition in Algorithmic Fairness and Explainability"

_entropy, 2023, doi:10.3390/e25050795_

Round 1
Reviewer 1 Report
The authors surveyed some recent and emerging applications of partial information decomposition in algorithmic fairness and explainability, which are of immense importance given the growing use of machine learning in high-stakes applications. The paper's organisation is well, and the manuscript requires some minor revisions for better quality.
- For reader's better understanding, give more information about Partial Information Decomposition (PID).
- Include the Fairness and Explainability metrics for better understating related to applications of partial information decomposition.
- Check the metrics of AIX360 (Explanability) and AIF (fairness) related to the model output and prediction.
- How to manage the Tradeoffs in Distributed Environments, and how it applicable in federated learning? Include more papers related to the tradeoff and Federated learning.
- For a better understanding spilt, the section 6 as future research directions and a summary of the contributions.
Author Response
We thank the reviewer for taking out the time to review this paper and for their comments and suggestions. We have prepared a revised version with our major edits colored in blue. Please find our responses to the review here:
The authors surveyed some recent and emerging applications of partial information decomposition in algorithmic fairness and explainability, which are of immense importance given the growing use of machine learning in high-stakes applications. The paper's organisation is well, and the manuscript requires some minor revisions for better quality.
RESPONSE: We thank the reviewer for their suggestions.
For reader's better understanding, give more information about Partial Information Decomposition (PID).
RESPONSE: We have now included some text in the introduction (Section 1) and in the section on background on partial information decomposition (Section 2) along with a new figure (See Figure 1). All the edits are colored in blue.
Include the Fairness and Explainability metrics for better understating related to applications of partial information decomposition. Check the metrics of AIX360 (Explanability) and AIF (fairness) related to the model output and prediction.
RESPONSE: We have made several edits in Sections 3 and 4 to address this suggestion (edits in blue). In particular, we have added two tables (Table 1 and Table 2) for a more accessible comparison with relevant metrics.
How to manage the Tradeoffs in Distributed Environments, and how it applicable in federated learning? Include more papers related to the tradeoff and Federated learning.
RESPONSE: We have made some changes in Section 5 to clarify the role of PID in fairness tradeoffs in federated learning. We note that this is a very recent area of research with very few prior works, and the tradeoff between local and global fairness has not received much attention.
For a better understanding spilt, the section 6 as future research directions and a summary of the contributions.
RESPONSE: We have addressed this comment.
Reviewer 2 Report
Paper surely deals with an interesting topic.
What is missing, in my opinion, is the relation with the other approached to fairness available in the literature:
- what are the adavantages of this approach
- what are the disadavantages of this approach
- what are the relations beween these and othere approaches
Moreover, as review paper, is too tight.
Author Response
We thank the reviewer for taking out the time to review this paper and for their comments and suggestions. We have prepared a revised version with our major edits colored in blue. Please find our responses to the review here:
Paper surely deals with an interesting topic.
What is missing, in my opinion, is the relation with the other approached to fairness available in the literature:
- what are the adavantages of this approach
- what are the disadavantages of this approach
- what are the relations beween these and othere approaches
Moreover, as review paper, is too tight.
RESPONSE: We thank the reviewer for this suggestion. We have made several edits in Section 3 and 4 to address this comment (edits in blue). In particular, we have added two new tables, Table 1 and 2 to provide a comparison with relevant measures. Wherever appropriate, we have also referred to the original papers ([8][31]) for a more elaborate discussion.
Reviewer 3 Report
The authors can elaborate on the significance of the research findings beyond the improved performance of the developed model. For instance, it could explore the potential impact of XAI on the field of cultural heritage and its preservation. The research does not provide any insight into the limitations of the research or the challenges encountered during the study, which could add valuable context to the findings.
The authors should provide quantitative and qualitative comparison with Decomposition in Algorithmic
Fairness and Explain ability.
Tabular comparison is missing for the interpretations of Explainable AI.
The authors can discuss topic with respect to dataset, model, tools like LIME and SHAPE.
Author Response
We thank the reviewer for taking out the time to review this paper and for their comments and suggestions. We have prepared a revised version with our major edits colored in blue. Please find our response to the review here:
The authors can elaborate on the significance of the research findings beyond the improved performance of the developed model. For instance, it could explore the potential impact of XAI on the field of cultural heritage and its preservation. The research does not provide any insight into the limitations of the research or the challenges encountered during the study, which could add valuable context to the findings. The authors should provide quantitative and qualitative comparison with Decomposition in Algorithmic Fairness and Explainability. Tabular comparison is missing for the interpretations of Explainable AI. The authors can discuss topic with respect to dataset, model, tools like LIME and SHAPE.
RESPONSE: We thank the reviewer for their review. We have made several edits in Section 3 and 4 to address some of the comments (edits in blue). In particular, we have added two new tables (Table 1 and Table 2) to discuss how these measures compare with relevant existing measures as well as the limitations of all the measures.
Since this is a review paper, we primarily provide a gentle introduction to the role of PID. We also refer to the original papers ([8],[31]) for experiments as well as a more detailed comparison to existing measures through extensive canonical examples which are out of the scope of this review paper.
Round 2
Reviewer 1 Report
The authors are addressed my comments.
Reviewer 2 Report
Authors have reasonably addressed reviewer comments.
Reviewer 3 Report
can be accepted